# Managing the Next Wave of Influenza and/or SARS-CoV-2 in the ICU—Practical Recommendations from an Expert Group for CAPA/IAPA Patients

**DOI:** 10.3390/jof9030312

**Published:** 2023-03-02

**Authors:** Jose Peral, Ángel Estella, Xavier Nuvials, Alejandro Rodríguez, Iratxe Seijas, Cruz Soriano, Borja Suberviola, Rafael Zaragoza

**Affiliations:** 1Intensive Care Unit, Hospital General Universitario Gregorio Marañón, 28007 Madrid, Spain; 2Intensive Care Unit, Medicine Department, University of Cádiz, University Hospital of Jerez, INIBiCA, 11407 Jerez, Spain; 3Intensive Care Unit, Hospital Universitari Vall d’Hebron, 08035 Barcelona, Spain; 4Intensive Care Department URV/IISPV/CIBERES, Hospital Universitari Joan XXIII, 43005 Tarragona, Spain; 5Intensive Care Department, Hospital Universitario de Cruces, 48903 Barakaldo, Spain; 6Department of Intensive Care Medicine and Intensive Care Unit, Hospital Ramón y Cajal, 28034 Madrid, Spain; 7Intensive Care Service, Hospital Universitario Marqués de Valdecilla, 39008 Santander, Spain; 8Intensive Care Service, Hospital Universitario Dr. Peset. Fundacion Micellium, C/La Armada Española, L’eliana, 46017 Valencia, Spain

**Keywords:** influenza-associated invasive aspergillosis, COVID-associated invasive aspergillosis, COVID-19, intensive care unit, critical patients

## Abstract

The aim of this study was to establish practical recommendations for the diagnosis and treatment of influenza-associated invasive aspergillosis (IAPA) based on the available evidence and experience acquired in the management of patients with COVID-19-associated pulmonary aspergillosis (CAPA). The CAPA/IAPA expert group defined 14 areas in which recommendations would be made. To search for evidence, the PICO strategy was used for both CAPA and IAPA in PubMed, using MeSH terms in combination with free text. Based on the results, each expert developed recommendations for two to three areas that they presented to the rest of the group in various meetings in order to reach consensus. As results, the practical recommendations for the management of CAPA/IAPA patients have been grouped into 12 sections. These recommendations are presented for both entities in the following situations: when to suspect fungal infection; what diagnostic methods are useful to diagnose these two entities; what treatment is recommended; what to do in case of resistance; drug interactions or determination of antifungal levels; how to monitor treatment effectiveness; what action to take in the event of treatment failure; the implications of concomitant corticosteroid administration; indications for the combined use of antifungals; when to withdraw treatment; what to do in case of positive cultures for *Aspergillus* spp. in a patient with severe viral pneumonia or Aspergillus colonization; and how to position antifungal prophylaxis in these patients. Available evidence to support the practical management of CAPA/IAPA patients is very scarce. Accumulated experience acquired in the management of CAPA patients can be very useful for the management of IAPA patients. The expert group presents eminently practical recommendations for the management of CAPA/IAPA patients.

## 1. Introduction

In December 2019, a pandemic caused by the SARS-CoV2 coronavirus spread rapidly from Asia to the rest of the world. Since then, it has been estimated that almost 200 million people have been infected worldwide, resulting in the deaths of several million people. This new infection was characterized by a high number of hospitalizations and admissions to the intensive care unit (ICU) in the pre-vaccination period, with most critically ill patients presenting with respiratory failure and the need for mechanical ventilation (MV).

In recent years, invasive pulmonary aspergillosis (IPA) in patients with influenza pneumonia has been recognized as an extremely serious emerging entity in critically ill patients, but evidence and recommendations regarding the management of these patients are scarce. Among patients with influenza who were admitted to the ICU and developed acute respiratory distress syndrome (ARDS), the incidence rates of influenza virus-associated IPA (IAPA) of up to 19% have been described. The mortality rate was 51% compared to 28% for influenza patients without IAPA [1].

The duration of stay in the ICU and of MV are both prolonged due to the extensive lung damage caused by both viruses and the high risk of superinfections. From a pathophysiological point of view, viral infections favor the development of fungal infection. The endothelial damage of the respiratory tract, the suppression of cellular immunity, the alteration of phagocytic activity, the dysfunction of mucociliary clearance, and the release of cytokines are just some potential explanations for the coexistence of viral and fungal infections [2].

The incidence of pulmonary aspergillosis in COVID-19 (CAPA) patients varies across published registries (UK, 14.1%; Italy, 27.7%; Germany, 26.3%; Netherlands, 19.4%; France, 19.6%) and is similar to the IAPA rate reported in ICU cohorts from Belgium and the Netherlands (19%). In the last 2 years, numerous studies have furthered our understanding of the behavior of the virus and investigated different treatment alternatives [1,3,4,5,6,7,8].

Recently published definitions of CAPA now enable the comparison of CAPA with IAPA patients. This will improve our still incomplete knowledge of the epidemiology of both entities and constitutes a first step in improving the management of superinfections associated with severe viral pneumonia in critically ill patients.

For this reason, a group of CAPA/IAPA experts was gathered to establish practical recommendations for the diagnosis and treatment of IAPA based on the available evidence and experience acquired in the management of CAPA patients. First, 14 areas were defined for which recommendations would be formulated. The PICO (patient, intervention, comparator, and outcome) methodology was used to search for information on both CAPA and IAPA in PubMed using MeSH terms in combination with free text. Based on the results, each expert developed recommendations for two to three aspects that they then presented to the rest of the group in various meetings in order to reach consensus.

The practical recommendations for the management of CAPA/IAPA patients have been grouped into 12 sections (Table 1).

## 2. Question 1: When Should CAPA/IAPA Be Suspected?

Recommendations:Include CAPA/IAPA in the differential diagnosis of respiratory superinfection in patients admitted to the ICU with viral pneumonia and associated clinical deterioration not explained by other factors. The presence of risk factors for fungal infection (EORTC/MSGERC host/risk criteria), prolonged or high-dose steroid treatment, prolonged mechanical ventilation, and/or the existence of structural lung injury should accentuate the need to assess the development of fungal infections.Consider the possibility of *Aspergillus* co-infection in patients admitted to the ICU with severe respiratory failure due to severe viral pneumonia caused by influenza/COVID-19.

The possibility of fungal infection should be considered at two key timepoints: co-infection at the moment of ICU admission and the initiation of mechanical ventilation; and early superinfection, in which case microbiological identification should be expected 5–7 days after ICU admission in patients with CAPA more related to immunoparalysis than to prolonged stay or the initiation of immunosuppressive therapy, generally in patients previously healthy lungs.

The characteristics of patients who present with severe COVID-19 pneumonia and experience complications due to the development of CAPA are highly compatible with those that account for the development of IAPA [9]. Published cases show that only a small proportion of patients meet the classical EORTC/MSGERC criteria [10]. Patients in both groups share similar characteristics, including high prevalence, the absence of classical host factors for invasive fungal infection (IFI), and the presence of lymphopenia. Risk factors for CAPA include immune dysregulation and immunoparalysis secondary to SARS-CoV-2 infection, poorly controlled diabetes mellitus (DM), previous lung disease, and therapies used for COVID-19, including corticosteroids and immunomodulators, such as tocilizumab [11]. However, there are some differences between CAPA and IAPA: for instance, *Aspergillus* tracheobronchitis and serum galactomannan (GM) positivity are less common in patients with CAPA than those with IAPA [8].

The limited use of bronchoscopies in critical COVID-19 patients during the first wave (the risk to healthcare personnel and the unprecedented volume of patients made routine bronchoscopy unfeasible) hindered the assessment of the diagnosis of patients with CAPA. Fifteen case series of critically ill patients with CAPA who underwent bronchoscopy (158 patients with CAPA out of 1702 COVID-19 patients; incidence, 9.3%) have been published, with most cases of CAPA considered probable/possible. The study by Bartoletti et al. incorporated bronchoscopy on days 0 and 7 of ICU admission and reported an incidence of 28% (30 out of 108 COVID-19 patients) and a high number of patients (14 of 108) with GM-positive bronchoalveolar lavage (BAL) samples upon ICU admission [4]. In their prospective study, Alanio et al. reported an incidence of CAPA of 33% among 27 COVID-19 patients who required mechanical ventilation [12]. Bronchoscopy was performed on day 3 post-intubation [12].

In published series, the time between diagnosis of influenza pneumonia and IAPA was short and almost always within 5 days (mean, 2 days). In the series involving CAPA patients, the time from COVID-19 onset to CAPA diagnosis ranged from 8–16 days and the period from intubation to CAPA diagnosis ranged from 3–8 days.

These data show that it is necessary to consider the possibility of co-infection upon ICU admission in patients with severe pneumonia due to influenza virus or COVID-19, and to establish the need for bronchoscopy as soon as possible. This is even more important before starting corticosteroid therapy in COVID-19 patients with respiratory worsening attributed to pulmonary fibrosis or organizing pneumonia [13,14,15].

## 3. Question 2: What Diagnostic Methods Should Be Used to Establish CAPA/IAPA Diagnosis and When Should They Be Applied?

(A)Microbiological Methods

Recommendations:Initiate a CAPA/IAPA diagnostic study in patients with severe viral pneumonia who are on MV and show clinical deterioration unexplained by other causes or in whom *Aspergillus* spp. is isolated in respiratory tract samples.Prioritize obtaining samples from the lower respiratory tract whenever possible, using a flexible bronchoscope to explore the airway and collect a BAL sample. Samples should be processed for culture in selective medium, GM determination, calcofluor and/or KOH staining, and *Aspergillus* PCR, depending on the availability of laboratory tests.The analysis of respiratory samples using lateral flow device (LFD) tests can be a valid alternative and is a useful tool for early diagnosis. It is recommended to use LFD tests in parallel with the aforementioned microbiological tests.The detection of *Aspergillus* spp. in respiratory samples not obtained by fiberoptic bronchoscopy (sputum, tracheal aspirate, non-bronchoscopic lavage) is insufficient for the diagnosis of CAPA/IAPA, except in cases of *Aspergillus* tracheobronchitis in IAPA, but does reiterate the need to initiate a diagnostic study using bronchoscopic samples.We do not recommend the systematic and routine screening of serum GM or 1-3-β-D-glucan in order to diagnose CAPA in critically ill patients with severe viral pneumonia: this is only recommended in the absence of other diagnostic options, in which case it should be performed serially, applying the usual cut-off points.We recommend evaluating a biopsy of the endobronchial mucosa when plaques, ulcerations, or other endobronchial lesions are observed when visualizing the trachea/bronchial tree with a fiberoptic bronchoscope in patients with suspected CAPA/IAPA, always taking into account the possibility of barotrauma associated with this technique.

Respiratory samples are preferred for the diagnosis of IFI. Initially, bronchoscopic techniques played a very minor role in the management of patients with severe pneumonia due to COVID-19. Fortunately, their use has progressively increased, and these techniques are now considered key tools for the diagnosis of CAPA/IAPA. In addition to allowing the direct inspection of the trachea and bronchi to identify lesions compatible with *Aspergillus* tracheobronchitis, bronchoscopy also enables the collection of BAL samples for microbiological analysis, which is fundamental in cases of suspected respiratory superinfection in patients with severe viral pneumonia and/or ARDS.

The presence of tracheobronchitis is more common in patients with IAPA, and has only been documented occasionally in CAPA. Endobronchial lesions that appear as pseudomembranes, ulcers, or eschars on bronchoscopy should be considered as evidence of CAPA, although these are less frequent in CAPA than IAPA patients, since SARS-CoV-2 only binds to ACE2 receptors, which are more abundant in the small-diameter airway [9].

For the diagnosis of CAPA/IAPA, BAL and lung biopsy are the techniques of choice, but biopsy is almost anecdotal, for obvious reasons. The detection of GM in BAL is highly indicative of IPA. A BAL sample that is GM-positive does not demonstrate tissue invasion: the probability of fungal infection increases if GM is also detected in serum. However, serum GM positivity in CAPA is infrequent: in some series, only 20% of patients had GM-positive serum, and cases of CAPA in patients with negative circulating GM have been described. This low sensitivity is reflected in most of published studies of non-neutropenic critical patients and is lower than the sensitivity reported in patients with IAPA (65%). Patients with CAPA present with fungal growth and invasion in the airway (in contrast to the rapid angioinvasion seen in neutropenic patients), and BAL samples are preferred for diagnosis [16].

LFD tests using blood and BAL samples have been used with good results for the diagnosis of IAPA. Although new data are needed to confirm the validity of LFD in CAPA, published data indicate similar results to those obtained with GM tests. The LFD test is simple, rapid, and can be performed in any microbiology laboratory [17].

In 2020, *Aspergillus* PCR was included in the guidelines for the definition of IFI, with two positive results required for diagnostic confirmation. Its validity in populations of patients with CAPA/IAPA is being studied, although results could be similar to those obtained in non-hematological populations [18].

We consider the detection of *Aspergillus* spp. in sputum, tracheal aspirate, or non-bronchoscopic lavage to be insufficient evidence for the diagnosis of CAPA, although positivity should emphasize the need for a deeper diagnostic study using BAL and fiberoptic bronchoscopy [19].

The use of another biomarker, (1-3) β-D-glucan, could be beneficial, but its utility in CAPA/IAPA is questionable, and we do not recommend it for screening patients with viral pneumonia, unless other microbiological tests are unavailable: owing to low specificity, it should always be performed serially, and other possible causes of false positives should be ruled out [2].

Some authors question whether, in the absence of other mycological evidence, a single positive GM or LFD result in serum is sufficient to classify a patient as having probable CAPA. It should be noted that the largest studies investigating the epidemiology of CAPA have unanimously reported higher mortality rates in patients with CAPA (52–71%) than those without (32–43%), especially when cultures and BAL GM are positive [7,20,21]. Therefore, high levels of alertness and clinical suspicion are necessary to ensure the early diagnosis and initiation of appropriate treatment. Owing to the lack of specificity of clinical and radiological data, mycological findings represent a basic pillar of diagnosis (Table 2).

Despite having a confirmed bacterial superinfection, the fact that patients with SARS-CoV-2 pneumonia have a high prevalence of IPA, for which influenza viral pneumonia has been identified as a risk factor, means it is advisable to rule out possible *Aspergillus* spp. infection. It is significant to contemplate the possibility of isolation of cryptic species in the etiology, as efforts for their identification beyond the genus level are important given their profile of resistance to antifungal therapy.
(B)Radiological Methods

Recommendations:Perform chest computed tomography (CT) in patients on MV with severe viral pneumonia and clinical deterioration not attributable to another cause.Initiate a CAPA/IAPA diagnostic study in patients with viral pneumonia on mechanical ventilation who present with new infiltrates, cavitated nodules, or halo sign on chest CT.

Some radiological findings of COVID-19 are similar to CAPA lesions and are not easy to differentiate, especially in patients with ARDS. Ghazafari et al. reported that there were no radiological differences between COVID-19 patients with or without associated fungal infection (105 patients) and that more than half had ground glass opacities or bilateral extensive consolidation [22]. Fortarezza et al. reported that the halo sign, typical of IPA in neutropenic patients, was also appreciable in critically ill patients with severe COVID-19 pneumonia without aspergillosis, explained by the presence of microthrombosis and vascular damage typical of COVID-19 [23].

We do not recommend systematically performing chest CT for the diagnosis of CAPA/IAPA, although for the study and management of complications it is more useful than looking for a nonexistent radiological pattern specific to CAPA/IAPA. Only in certain patients with angioinvasive forms or nodule or cavern formation could chest CT provide relevant information.

## 4. Question 3: What Approach Should Be Taken When Certain Diagnostic Methods Are Not Possible?

Recommendations:Prioritize bronchoscopic BAL samples followed by a blind mini-BAL and by non-invasive tests (tracheal aspirate) if fiberoptic bronchoscopy cannot be performed.Perform the diagnostic tests available when CAPA/IAPA is suspected, despite the scarcity of evidence supporting other tests of lesser diagnostic value.If clinical suspicion is high, and results are either inconclusive or will not be available for some time, we recommend starting antifungal treatment, which can then be discontinued, modified, or continued depending on the results.

The type of sample obtained is very important to maximize diagnostic accuracy: bronchoscopic BAL samples are the sample of choice, followed by a blind mini-BAL and, finally, non-invasive tests (bronchial aspirate) (Table 3).

In case of suspected CAPA/IAPA, available diagnostic tests should be performed: fungal culture with the possibility of an antifungigram, the determination of GM in a respiratory sample, LFD, real-time PCR for *Aspergillus*, Kohl and calcofluor staining (always performed by experts, as it is an insensitive technique that is highly dependent on experience). In certain situations, given the potential non-availability of tests with the highest diagnostic yield, we recommend using whatever tests are available. However, it should be borne in mind that very strict cut-off points cannot be applied, owing to the particular vulnerability of the population in question. Carrying out two consecutive tests using serum or non-invasive samples will improve diagnostic capacity.

As already described in patients with extensive pneumonia due to COVID-19, it is difficult to distinguish between co-infection, superinfection, and *Aspergillus* colonization. While recently published ECMM/ISHAM diagnostic criteria for CAPA consider samples obtained by non-bronchoscopic lavage as valid in the case of possible CAPA, we believe that other microbiological samples are necessary and that clinical and radiological evolution should be monitored [19]. A blind non-bronchoscopic lavage sample can be obtained using a closed technique, which decreases the risk of transmission.

## 5. Question 4: What Is the Recommended Antifungal Treatment in CAPA/IAPA?

Recommendations:Currently, there is no scientific evidence to support the superiority of one antifungal over another for the treatment of CAPA/IAPA, nor are there data that suggest that treatment should differ to that received by other patients with IPA. Therefore, it is recommended to follow the treatment indications in current national and international guidelines [10,19], taking into account the peculiarities of critical patients and, in particular, of patients with severe viral pneumonia due to influenza or COVID-19.It is recommended to include isavuconazole or liposomal amphotericin B as first-line drugs for the treatment of CAPA/IAPA patients.The antifungal treatment of CAPA/IAPA patients is recommended until diagnosis is confirmed.

There is no controversy regarding the indication of antifungal treatment in patients with proven or probable CAPA/IAPA.

The diagnosis of possible CAPA/IAPA is based on the existence of radiological alterations and microbiological findings in a respiratory sample not obtained by bronchoscopy, indicating either contamination or colonization. However, given that delayed treatment in CAPA/IAPA patients is associated with a significant increase in mortality [20,24], it is recommended to start antifungal treatment as soon as suspicion is established and to reassess once microbiological test results are obtained [19].

In critically ill patients, pathophysiological alterations can generate a greater risk of variability in the behavior of antifungal drugs. Voriconazole is not considered a first-line drug in this type of patient, owing to its narrow therapeutic window and interactions. Because it is metabolized via CYP2C19, CYP2C9, and CYP3A4, voriconazole can potentially interact with drugs commonly used in COVID-19, such as remdesivir [19] and dexamethasone [25]. Dexamethasone works by inducing the various enzymes of the CYP450 system, thereby decreasing voriconazole levels [26], while remdesivir competes with voriconazole as a substrate for the CYP3A4 enzyme, altering voriconazole levels [19]. In the treatment of IAPA, the most common pharmacological interaction occurs in cases of the concomitant administration of voriconazole and cloxacillin, which is frequently used to treat co-infections caused by *S. aureus*. Although the underlying mechanism is not fully understood, the use of cloxacillin is associated with extremely low levels of voriconazole [8].

It is important to note that isavuconazole is metabolized via CYP3A4 and may also cause interactions. However, such interactions are generally less pronounced than those observed for voriconazole [19].

For the aforementioned reasons, isavuconazole or liposomal amphotericin B are recommended as first-line drugs for the treatment of CAPA/IAPA patients.

We consider it necessary to establish a subgroup of patients who could benefit from the early initiation of antifungal treatment: patients with a high suspicion of prediagnostic microbiological CAPA/IAPA, i.e., those in whom clinical suspicion is high even in the absence of compatible microbiological data (either because it is not possible to perform the necessary tests or because the results are not yet available); in an environment with high CAPA/IAPA prevalence; and with clinical deterioration not explained by any other cause. We classify these patients using the acronym SPAOT (suspected pulmonary aspergillosis on treatment).

## 6. Question 5: Which Antifungal Treatment Is the Most Suitable if Resistance-Related Problems Arise? When Should We Suspect Resistance in CAPA/IAPA?

Recommendations:In cases of documented resistance to azoles, we recommend treatment with liposomal amphotericin B.There are no specific factors described in the literature to indicate suspected azole resistance in CAPA/IAPA, nor is there evidence to suggest that the risk of resistant *Aspergillus* infections differs from that of other ICU populations. Suspicion should be based on local susceptibility studies.In cases of *Aspergillus* infections in centers with a high prevalence of azole-resistant isolates (>10%), we recommend initial treatment with liposomal amphotericin B until antifungal susceptibility results are available.

Acquired resistance in *Aspergillus fumigatus* sensu stricto and other *A. fumigatus* complexes usually develops following prolonged exposure to antifungals (either as treatment or as a consequence of environmental exposure) and is usually the result of point mutations in the *Aspergillus* gene and/or the insertion of tandem repeats. Resistance can arise de novo or can occur during the initiation of treatment [27]. Since respiratory culture samples can simultaneously contain both azole-susceptible and -resistant isolates, a minimum of five colonies should be tested [28].

The prevalence of azole-resistant isolates varies by geographic region. A multicenter study conducted in 19 European countries reported an overall rate of azole resistance of 3.2% (0–26%): the most frequently observed mutation was TR34/L98H [29]. A review of the literature reveals azole resistance rates of 2–12% in clinical samples (Brazil, 3.5%; China, 5.8%; Japan, 6.1%; Pakistan, 6.6%; USA, 0.6–11.8%) and even higher rates in environmental samples (Tanzania, 13.9%; Colombia, 9.3%) [30].

If an *Aspergillus*-positive culture is obtained, an antifungal susceptibility study could be considered. This is advisable in geographic areas with a documented high prevalence of antifungal resistance (>10%) in cases of therapeutic failure and in patients with previous exposure to azoles. Owing to the costs and difficulties associated with culture studies, the current rates of azole resistance may be underestimated.

Although cases of CAPA/IAPA caused by azole-resistant *Aspergillus* have been documented, there are no data to suggest that resistance rates in these patients are higher than those of critically ill patients in general [10]. To date, five cases of azole-resistant CAPA have been described. In four cases the TR34/L98H mutation was identified, and in one case the TR46/Y121F/T289A mutation was found, which is associated with environmental selection [31,32,33,34]. There are no specific factors described in the literature to suspect azole resistance in critically ill CAPA/IAPA patients, nor are there any indications that the risk of resistant *Aspergillus* infections differs from those of other ICU populations. Therefore, we recommend that suspicion be based on local sensitivity studies.

Some studies have shown that azole resistance is associated with therapeutic failure if azole treatment is maintained over time [35]. For this reason, liposomal amphotericin B is recommended as first-line treatment in patients with infections caused by azole-resistant *Aspergillus*, and in centers with an incidence of azole-resistant isolates >10%. Another alternative proposed by some authors is combined therapy with azoles and echinocandins, although in our opinion, this combination should only be used when the use of amphotericin B lipid formulations is formally contraindicated [36].

## 7. Question 6: In What Situations Should We Monitor Whether Treatment Is Appropriate or not?

Recommendations:Critical CAPA/IAPA patients treated with voriconazole should undergo drug level monitoring at least weekly, and we recommended doing so twice during the first week of treatment. Patients receiving corticosteroids and those undergoing extrarenal clearance and/or ECMO therapies benefit most from routine monitoring.In critically ill CAPA/IAPA patients treated with isavuconazole who have high BMI and/or are undergoing renal replacement techniques, drug level monitoring can be considered if the necessary techniques are available.The monitoring of triazole levels should be performed in real time.We recommend using liposomal amphotericin B for the antifungal treatment of CAPA/IAPA patients if the determination of azole levels is not possible or if there is a high risk of drug interactions.

Therapeutic drug monitoring is a proven effective tool to improve clinical outcomes in patients, minimizing toxicity and maximizing therapeutic effectiveness [19].

The following are the main clinical scenarios in which drug level monitoring should be considered: starting or discontinuing drugs that can have pharmacological interactions with the antifungal; dose change; when the response to treatment is not as expected, due to lack of efficacy or suspected toxicity; patients receiving support treatments and undergoing depurative membrane or extracorporeal membrane oxygenation; patients who have organic dysfunctions and require treatment adjustments; and high-risk patients (advanced age, COPD and serum GM > 0.5, high severity score on admission, and patients receiving renal replacement therapies) [2].

In the treatment of CAPA/IAPA, we recommend monitoring levels of voriconazole and/or posaconazole in real time. Drug interactions, pathophysiological changes due to renal and/or hepatic failure, malnutrition, alterations in protein binding, vasopressors, renal replacement techniques, and ECMO further increase this variability and may interfere with drug absorption, distribution, metabolism, and excretion [19].

This is especially important in patients with COVID-19 pneumonia receiving dexamethasone and remdesivir [26,37]. For this reason, we recommend weekly monitoring of levels (initially, twice per week). Although there are some discrepancies between centers, voriconazole and posaconazole concentrations of 2–6 mg/L and 1–3.75 mg/L, respectively, seem optimal [28].

Some articles have reported that in patients with CAPA voriconazole treatment for up to 7 days may be necessary to reach the therapeutic range; in one series more than 80% of patients with CAPA presented subtherapeutic levels on day 5 of treatment [2]. Therefore, it is recommended to use voriconazole only in centers in which it is possible to obtain drug level data on the same day as blood extraction. Moreover, given the high risk of suboptimal voriconazole levels in the days after beginning treatment, we recommend considering the addition of a second drug with activity against *Aspergillus*, preferably liposomal amphotericin B, until stable therapeutic levels are reached.

Initially, Troutine monitoring of isavuconazole levels was not considered, based on data from a pivotal study that found no association between blood levels and drug efficacy and/or toxicity. An analysis of eight studies in which isavuconazole levels were monitored in a total of 368 patients produced no conclusive findings in this regard [38]. However, suboptimal isavuconazole concentrations have been reported in critically ill patients, and are more frequent in obese patients and those undergoing renal replacement techniques [19]. Höhl et al. determined isavuconazole levels in 41 critical patients, and found that up to 31.7% of measurements taken in the trough period were below the limit of 1 µg/mL (EUCAST breakpoint), and 12.1% of peak levels did not reach 1 μg/mL [39]. Significantly lower levels were associated with a higher BMI and a higher SOFA score, which may suggest the need to monitor levels in this specific population. At this point in time, we cannot definitively recommend monitoring, although this would be reasonable in these patient subgroups, assuming the necessary techniques are available.

## 8. Question 7: What Procedure Should Be Followed in the Event of Treatment Failure?

Recommendations:Reconsider the diagnosis and/or antifungal treatment used if a lack of microbiological eradication or an inadequate spectrum are suspected.Reconsider the antifungal therapy in cases of pharmacological failure: pharmacokinetics (PK)/pharmacodynamics (PD), drug–drug interaction (DDI), and therapeutic drug monitoring (TDM) is suspected.Modify and individualize treatment if therapeutic failure is thought to be linked to patient-associated factors (risk stratification, clinical profile, the control of the infectious focus, etc.).

A CAPA/IAPA patient is considered refractory when the clinical, microbiological, and/or radiological data do not improve despite continued treatment. Continuous clinical assessment is required, including GM determination, a repeated bronchoscopy, and monitoring of GM levels, if the patient shows an unfavorable course. It is important to rule out the possibility that clinical deterioration is due to immune reconstitution.

The management of refractory CAPA/IAPA is complex but should always include a review of factors that could account for therapeutic failure, such as suboptimal drug levels, the progression of the underlying pathology, diagnostic error, and/or new concomitant infection. The control of the source of infection and antifungigram findings should be taken into account. Echinocandins are considered second-line treatments. A good level of evidence supports the use of liposomal amphotericin B and voriconazole as rescue therapy in IPA, and there is some evidence supporting posaconazole as rescue therapy in patients with voriconazole failure. The use of isavuconazole as a rescue therapy has not been reported but would appear to be a feasible option. Combination therapy has not been studied as a rescue therapy in CAPA/IAPA patients [40,41,42,43,44,45].

## 9. Question 8: In CAPA/IAPA Patients Receiving Antifungal Treatment, Is the Administration of Corticoids Associated with Higher Mortality and/or Higher Incidence of MV-Associated Pneumonia?

Recommendations:Corticosteroid therapy in influenza pneumonia is associated with increased mortality and therefore should not be used. While we have found no studies on the effect of corticosteroid therapy in patients with IAPA, its immunosuppressive effect could worsen the prognosis in these patients.Corticosteroid therapy in influenza pneumonia is associated with a higher incidence of superinfections, and therefore corticosteroids should not be administered as adjuvant therapy. We have found no studies on the incidence of superinfections in patients with IAPA, but the effect observed in influenza patients can be extrapolated to patients with IAPA, and therefore corticosteroids should be avoided.Corticosteroid therapy in COVID-19 pneumonia may be associated with lower mortality. We have found no evidence of the impact of corticosteroid treatment in patients with CAPA, and therefore can make no recommendation in this regard.Corticosteroid therapy in CAPA should be avoided owing to the apparent increased likelihood of superinfection, although supporting evidence is very scarce.

### 9.1. IAPA and Mortality 

In viral pneumonia, corticosteroid treatment has not shown beneficial effects in patients with MERS or SARS infection. However, during the influenza A (H1N1) pandemic, corticosteroid therapy was proposed as a potential adjuvant treatment.

Although the studies by Waldeck [46] and Martín-Loeches [47] reported no increase in mortality, Moreno et al. [48] and Chen et al. [49] described increased mortality in patients treated with corticosteroids. In agreement with these observations, the findings of several meta-analyses suggest that corticosteroid administration in patients with influenza A is associated with increased mortality [50,51]. Therefore, corticosteroids should not be used as an adjuvant treatment in influenza pneumonia.

We have found no studies comparing IAPA patients treated with versus without corticosteroids, and therefore can make no recommendations based on direct evidence. Evidence from other observational studies indicates a greater predisposition to IAPA in patients receiving corticosteroids, as well as a less favorable clinical course [52]. Therefore, the recommendation not to administer corticosteroids in patients with severe pneumonia due to influenza should also be applied to patients with IAPA (Table 4).

### 9.2. IAPA and VAP Incidence 

Corticosteroid therapy in influenza has been proposed to attenuate the marked and sustained lung inflammation in patients with viral pneumonia. However, its immunosuppressive effect may lead to a greater risk of superinfections. Observational studies and various meta-analyses suggest that corticosteroid therapy is associated with a higher rate of superinfections, which in turn is associated with higher mortality. According to the evidence, this higher incidence of superinfection includes those caused by bacteria and *Aspergillus* spp., and therefore corticosteroid therapy should be avoided in these patients [1,52] (Table 4).

### 9.3. CAPA and Mortality 

During the COVID-19 pandemic, the RECOVERY study (*n* = 6000 patients) demonstrated that dexamethasone treatment (6 mg) was associated with lower mortality, an effect that was more pronounced in patients requiring MV (RR 0.65; 95% CI 0.48–0.88). This resulted in a change in clinical practice, with dexamethasone treatment being used more generally to treat COVID-19 patients with respiratory failure. The REMAP-CAP study showed that tocilizumab or sarilumab was able to reduce mortality when started early in critically ill patients (in-hospital mortality: tocilizumab group, 28%; sarilumab group, 22%; control group, 35.8%). Similarly, the RECOVERY study confirmed an improvement in survival in patients treated with tocilizumab + dexamethasone [53,54].

However, there is no evidence as to the effects of corticosteroids in patients with CAPA. Although no data support an increased incidence of IPA in patients receiving immunotherapy, the difficulty in diagnosing CAPA and the suboptimal recording of fungal infections in these studies hinder the understanding of the impact of these treatments in CAPA patients. We cannot establish a clear recommendation in this regard. Clinicians should be aware that CAPA as a complication of COVID-19 in critically ill patients may be favored by these immunosuppressive treatments, which could suppress the host’s antifungal defense. However, this interference with the immune response has shown beneficial effects in critically ill patients with COVID-19 and could reduce the risk of IPA by limiting epithelial and tissue damage. Currently, available data do not support the discontinuation of dexamethasone treatment in cases of CAPA diagnosis [55].

In patients with COVID-19 who develop ARDS or bronchiolitis obliterans organizing pneumonia (BOOP) and receive corticosteroid therapy, the need to continue treatment should be assessed in the event of development of a fungal infection and CAPA [56] (Table 5).

### 9.4. CAPA and VAP Incidence

Evidence is lacking in this area. However, in contrast to the findings of Moreno et al., who found that corticosteroid administration was not associated with an increased risk of VAP [57], Leistner et al. [58], Lee et al. [59] and Søvik et al. [60] reported a higher incidence of superinfections and CAPA associated with dexamethasone treatment. Despite the paucity of evidence, the extrapolation of findings in patients with other respiratory infections indicates that corticosteroid treatment in patients with COVID-19 and CAPA should be reconsidered given the possibility of increased immunosuppression (Table 5).

## 10. Question 9: In Which CAPA/IAPA Patients Is Combined Antifungal Therapy Associated with Lower Mortality and/or a Shorter Hospital Stay?

Recommendations:It is not possible to make a definitive recommendation for CAPA/IAPA patients, owing to the scarcity of published data. However, based on expert opinion, we recommend combination therapy with liposomal amphotericin B in critically ill patients with a poor clinical course until antifungal levels are within the therapeutic range, with the option to later de-escalate to monotherapy.

Evidence supporting combination therapy as the first line of treatment in patients with IPA is weak, although it can be considered in critically ill patients with severe infection or when azole resistance is suspected. Combining antifungals is generally conditioned by expert opinions or individual preferences, or used as rescue treatment for refractory infections. The largest relevant clinical trial is a multicenter study of 459 patients that compared the combination of voriconazole + anidulafungin with voriconazole monotherapy. No differences in 6-week mortality (19.3% vs. 27.5%, *p* = 0.09), 12-week mortality, or IPA-associated mortality were observed in patients who received combination therapy. A post hoc analysis showed that the group with positive serum GM had significantly lower mortality (15.7% vs. 27.3%; *p* = 0.037), suggesting a certain benefit in those patients diagnosed early [28,61].

Other international guidelines also suggest a role for initial combination therapy in the very severe forms of the disease, but based on the strength of the evidence, combination therapy as the first line of treatment in IPA can only have a weak recommendation, and the results cannot be extrapolated to CAPA/IAPA patients as relevant data are lacking [62].

## 11. Question 10: When Should Antifungal Treatment Be Withdrawn in CAPA/IAPA Patients?

Recommendations:There is no evidence to support recommendations on when to withdraw antifungal treatment in CAPA/IAPA patients. In our opinion, the duration should be based on the clinical response and patient’s immune status and should be between 4 and 6 weeks in patients with an adequate clinical course. In those with an unfavorable course, antifungal treatment should be maintained while other diagnoses are ruled out and microbiological tests are repeated, including BAL GM and an antifungigram.

This aspect has long been a source of controversy, as demonstrated by the disparate criteria in the American versus European guidelines. While the IDSA recommends continuing antifungal therapy for a minimum of 6–12 weeks, with duration based on the severity of infection, the persistence of immunosuppression, and response to treatment (very much oriented toward hematology patients), the European ESCMID/ECMM guidelines recommend adjusting the treatment duration based on efficacy and clinical response [28,61]. However, no clinical trials have assessed the time required to achieve cure in CAPA/IAPA patients. We consider 4–6 weeks to be an appropriate duration of treatment, but in patients with poor clinical course treatment should be continued while diagnoses are reconsidered (to rule out other etiologies) and tests should be repeated, including BAL GM and an antifungigram. The serialization of serum GM tests does not provide any added value, owing to their low sensitivity in these patients.

## 12. Question 11: What Is the Management Approach for Patients with Viral Pneumonia Caused by Influenza/COVID-19 with a Positive Culture for Aspergillus?

Recommendations:There is no scientific evidence to support specific recommendations when a positive *Aspergillus* culture is obtained in a patient with severe viral pneumonia due to influenza/COVID-19.In our opinion, management should be individualized and the patient’s clinical status assessed. If severe respiratory compromise is observed, appropriate antifungal treatment should be started early, and a comprehensive microbiological study carried out simultaneously to assess the relevance of microbiological positivity. On the other hand, if the positive culture is obtained during the phase of MV withdrawal, when the need for oxygen is reduced or even before invasive ventilatory support has been withdrawn and, therefore, the patient’s levels of respiratory failure is not considered severe, we consider this evidence of colonization, which does not require specific treatment.The establishment of antifungal treatment in patients with severe viral pneumonia and *Aspergillus* colonization is controversial, and no specific recommendations can be made in this regard.

There is no clear evidence on which to base recommendations about when to treat patients with a positive *Aspergillus* culture with severe viral pneumonia due to influenza/COVID-19. Most relevant published reports are case series, in which the diagnosis is open to debate.

The difficulty in distinguishing between *Aspergillus* colonization and fungal infection is well described and constitutes an important challenge: in studies of ICU patients with CAPA, those with a positive *Aspergillus* culture showed higher mortality, although this difference was not statistically significant in all studies [10]. In the study by Bartoletti et al., of thirty patients with CAPA, sixteen received antifungal treatment and fourteen did not (seven due to postmortem diagnosis and seven due to a clinical decision). Survival among those treated with voriconazole was 54% versus 41% among those who did not receive voriconazole [4]. In the series by White et al., overall mortality was 46.7% in patients with CAPA who received appropriate antifungals compared with 100% in patients who did not [3].

Case series have described patients diagnosed with CAPA who survived without receiving antifungal treatment. In the series by Alanio et al., of seven patients with possible/probable CAPA who did not receive antifungals, five survived [12]. Survival in this case may be due to several factors, the most important of which is the absence of invasive infection and the presence of mere colonization. These observations could imply that in some critically ill patients with COVID-19 pneumonia, *Aspergillus* positivity may reflect colonization, the impact of which in patients with severe lung involvement by SARS-CoV2 should be properly studied [63,64].

In cases of a positive *Aspergillus* culture in patients with severe viral pneumonia due to influenza/COVID-19, an individualized approach should be taken, first considering whether the patient’s situation is critical and involves severe respiratory compromise, and next starting appropriate early antifungal treatment, while simultaneously carrying out a comprehensive microbiological study to assess the importance of microbiological positivity. On the other hand, if a positive culture is obtained in a critically ill patient with viral pneumonia who is in the phase of withdrawal from mechanical ventilation, has lower oxygen requirements, or has even undergone the withdrawal of invasive ventilatory support and, therefore, does not fulfill the criteria for severe respiratory failure, we recommend classification as colonization, which does not require specific treatment. In such cases, it is advisable to further pursue the diagnosis and analyze a future sample. If subsequent respiratory deterioration occurs, we recommend collecting new microbiological samples and initiating early treatment, since the influenza virus and SARS-CoV2 are risk factors for the development of IPA.

In a clinical scenario in which *Aspergillus* colonization occurs, antifungal treatment with the goal of achieving microbiological eradication is plausible and could be justified owing to the potential complications in patients with severely damaged lung tissue. There are no relevant scientific data, and the CAPA/IAPA working group has not reached a consensus based on which recommendations could be established. Although experiences in transplant programs have been reported, the value of extrapolating those results to these patients is questionable. The options considered by the working group range from maintaining an expectant and vigilant attitude and only treating in case of deterioration to treating only high-risk patients, in whom treatment with nebulized antifungals could be considered.

## 13. Question 12: In What Type of Patients Could Antifungal Prophylaxis Be Recommended?

Recommendations:Antifungal prophylaxis is not recommended in patients with severe influenza virus/COVID-19 pneumonia undergoing MV.Early screening and antifungal treatment for *Aspergillus* should be started in patients with a clinical suspicion of fungal infection, risk factors for CAPA/IAPA, colonization, and poor clinical course. The recommended options are azoles or liposomal amphotericin B, depending on the prevalence of azole resistance.

New studies are needed to establish the need and efficacy of CAPA/IAPA prophylaxis in critically ill patients who do not have a hematological malignancy and are not transplant recipients.

In an observational study of the efficacy of CAPA prophylaxis, antifungal prophylaxis (98% with posaconazole) was administered to 75 patients from a series of 132 patients critically ill with COVID-19 (57%). Of the ten patients diagnosed with CAPA, nine were in the group that did not receive prophylaxis. However, there was no difference in 30-day mortality between the two groups (37% in both) [65]. In the POSA-FLU study, Vanderberke et al. analyzed the efficacy of posaconazole prophylaxis in a small population of patients critically ill with severe influenza viral pneumonia. This study, which had an insufficiently well-conducted methodology, reported no decrease in overall mortality and no reduction in the incidence of IAPA [66].

Antifungal prophylaxis is a cornerstone of the management of patients at high risk of invasive fungal infection, such as myeloid leukemia or transplant patients. However, to date, there are no data to support the initiation of prophylaxis in patients with severe viral pneumonia.

## 14. Conclusions

Unfortunately, patients with CAPA continue to pose a challenge for clinicians, particularly in the context of the current pandemic. Despite numerous papers published in recent years, available evidence supporting practical recommendations for the management of CAPA/IAPA patients is very scarce and many issues remain to be clarified. In order to optimize management and improve patient outcomes, a better, evidence-based diagnostic approach is essential to help guide decision-making by physicians, which currently ranges from early treatment to a “wait-and-see” approach.

Accumulated experience in the management of patients with CAPA suggests that the appropriate approach should be based on early treatment and, in the absence of confirmation, the safe de-escalation of the established treatment. The duration of antifungal treatment should also be reviewed: in the absence of prolonged neutropenia, a shorter treatment duration appears feasible, although further studies are needed to confirm this hypothesis. Finally, additional studies are needed to investigate the potential benefit of antifungal prophylaxis, particularly the effect of nebulized antifungals on the incidence and mortality of CAPA/IAPA. The resulting evidence could prove highly valuable to better understand the role of CAPA/IAPA in the mortality of critically ill patients with severe viral pneumonia.

## Figures and Tables

**Table 1 jof-09-00312-t001:** Key Questions.

1	When should CAPA/IAPA be suspected?
2	What diagnostic methods should be used to establish CAPA/IAPA diagnosis and when should they be applied?
3	What to do when it is impossible to use certain diagnostic methods? What approach should be taken when certain diagnostic methods are not possible?
4	What is the recommended antifungal treatment in CAPA/IAPA?
5	Which antifungal treatment is the most suitable if resistance-related problems arise? When should we suspect resistance in CAPA/IAPA?
6	In what situations should we monitor whether treatment is appropriate or Not?
7	How should treatment failure be defined and what procedure should be followed if this occurs?
8	In CAPA/IAPA patients, is the administration of corticosteroids associated with increased mortality and/or increased incidence of pneumonia associated with mechanical ventilation?
9	In CAPA/IAPA patients, is combined antifungal therapy associated with lower mortality and/or a shorter hospital stay? In which CAPA/IAPA patients is combined antifungal therapy associated with lower mortality and/or shorter hospital stay?
10	When should antifungal treatment be withdrawn in CAPA/IAPA patients?
11	What is the management approach for patients with viral pneumonia caused by influenza/COVID-19 with a positive culture for *Aspergillus*?
12	In what type of patients could antifungal prophylaxis be recommended? In what type of patients with severe viral pneumonia could antifungal prophylaxis be recommended?

**Table 2 jof-09-00312-t002:** Validation of Mycological Tests in Multi-Center Studies of CAPA Patients [22,23,24].

Diagnostic Test	Sensitivity	Specificity
GM in BAL > 1	74% (77/104)	99% (268/272)
BAL culture	53% (56/106)	100% (298/298)
Lateral flow BAL	52% (15/29)	98% (60/61)
BAL PCR	42% (48/115)	100% (49/49)
Serum GM > 0.5	19% (20/106)	100% (379/380)
1-3-β-D-glucan	38% (8/21)	85% (29/34)

BAL, bronchoalveolar lavage; GM, galactomannan.

**Table 3 jof-09-00312-t003:** Diagnostic Methods Used for the Diagnosis of CAPA/IAPA. Advantages and Disadvantages.

Diagnostic Test	Comments for CAPA Patients	Advantages	Disadvantages
Lung biopsy	Post-mortem CT-guided biopsies have been used as an alternative to necropsy	Provides definitive CAPA/IAPA diagnosis	High risk of complications
BAL with FBC	In the first wave, practically dismissed due to risk to health personnel. Rarely used during the first wave due to risk to health personnel.	Visualization of trachea and bronchi BAL GM, LFD, and PCR Directed sample	Aerosol generation Potentially poor tolerance in some patients
Not bronchoscopic lavage Non-bronchoscopic lavage	Proposed as an alternative to BAL	Obtains sample from lower respiratory tract. Validated technique for VAP Closed technique	Not well validated for CAPA/IAPA Not well validated for GM/PCR Blind sample
Tracheal aspirate	Colonization in patients with COVID-19?	Easy to perform in MV patients	Sample is less representative of the lower respiratory tract Not validated for biomarkers
Sputum	Colonization in patients with COVID-19?	Easy to perform on all patients	Sample is less representative of the lower respiratory tract Not validated for biomarkers
Serum	Frequently negative in CAPA	Allows GM, LFD, 1-3-β-D-glucan, and PCR Easily obtained	1-3-β-D-glucan results not specific

BAL, bronchoalveolar lavage; GM, galactomannan; LFD, lateral flow device test; VAP, ventilator-associated pneumonia; MV, mechanical ventilation; PCR, polymerase chain reaction.

**Table 4 jof-09-00312-t004:** Studies Assessing the Administration of Corticosteroids in Patients with IAPA.

Assessment	Type/Number of Patients	Conclusion	Cort. vs. Non-Cort. Mortality
Chen et al. (2020) [49]	Retrospective/693 patients	Increase in mortality	
Zhang et al. (2020) [51] Waldeck F.)	Retrospective/81 flu patients	No increase in mortality	OR = 2.2 (0.3–12)
Lansbury et al. (2020) [52]	Meta-analysis/21 studies (1 RCT)	Increase in mortality with corticosteroids Increase in nosocomial infection	OR = 3.90 (2.31–6.60) 15 studies; aHR 1.49 (1.09–2.02) 6 studies OR = 2.74 (1.51–4.95)
Yang JW et al. (2015) [53]	Meta-analysis/19 studies (4916 patients)	Increase in mortality with corticosteroids Increases in number of days on MVIncrease stay at ICU	OR = 3.16 (2.09–4.78) WMD = 3.82 (1.49–6.15) WMD = 4.78 (2.27–7.29)
Moreno G et al. (2018) [54]	Retrospective PS/1846 patients	Increase in mortality	HR = 1.32 (1.08–1.60)
Zhang Y et al. (2015) [55]	Meta-analysis/6105 patients	Increase in mortality (cohort studies) Increase in mortality (case–control)	RR = 1.85 (1.46–2.33) RR = 4.22 (3.10–5.76)
Schauwvlieghe et al. (2018) [56]	Retrospective/432 patients	Increased risk of IAPA	aOR = 1.59 (1.30–1.99)
Lansbury L et al. (2019) [57]	Meta-analysis/30 studies (1 RCT) 99,224 patients	Increase in nosocomial infection	OR = 3.90 (2.31–6.60) OR = 2.74 (1.51–4.95)
Martin-Loeches I. (Leistner et al. (2011) [58]	Prospective/220 patients	No increase in mortality Increased incidence of pneumonia	HR = 1.3 (0.7–2.4) OR = 2.2 (1.0–4.8)
Lee N et al. (2015) [59]	Retrospective/2649 patients	Increase in mortality Increase in superinfection	aHR = 1.73 (1.14–2.62) 9.7% vs. 2.7%; *p* < 0.001

HR, hazard ratio; IAPA, ICU, intensive care unit; MV, mechanical ventilation; OR, odds ratio; RCT, randomized controlled trial; WMD, weighted mean difference.

**Table 5 jof-09-00312-t005:** Studies Assessing the Administration of Corticosteroids in Patients with CAPA.

Assessment	Type/Number of Patients	Conclusion	Cort. vs. Non-Cort. Mortality
Cochrane (2021) [53]	Meta-analysis/7989 patients	Probable decrease in mortality	RR = 0.89 (0.80–1.00)
Li H et al. (2021) [54]	Meta-analysis/6772 patients	Decrease in mortality	OR = 0.70 (0.54–0.92)
Chong et al. (2022) [55]	Meta-analysis/729 patients	No increase in mortality	OR = 0.69 (0.19–2.58)
Chaharom et al. (2022) [56]	Meta-analysis/18,190 patients	No overall decrease in mortality Decrease in mortality in RCT	OR = 1.12 (0.83–1.50) OR = 0.80 (0.73–0.88)
Moreno et al. (2021) [57]	Retrospective/1853 ventilated patients	Increase in mortality > 17 days No increase in VAP	HR = 0.53 (0.39–0.72) HR = 1.68 (1.16–2.45) OR = 1.05 (0.83–1.34)
Leistner et al. (2022) [58]	Retrospective/529 patients	Increased risk of CAPA	OR = 3.11 (1.11–8.69)
Søvik et al. (2022) [60]	Prospective/156 patients	Increased risk of superinfection	OR = 3.7 (1.80–7.61)

HR, hazard ratio; IAPA, OR, odds ratio; RCT, randomized controlled trial; VAP, ventilator-associated pneumonia.

## Data Availability

Not applicable.

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
