# Peer review of "Managing the Next Wave of Influenza and/or SARS-CoV-2 in the ICU—Practical Recommendations from an Expert Group for CAPA/IAPA Patients"

_jof, 2023, doi:10.3390/jof9030312_

Round 1
Reviewer 1 Report
It is a review study of publications on invasive pulmonary aspergillosis in patients with viral pneumonias due to Covid 19.
It is not a study that provides original data except for the authors' comments in different items. It has a practical interest because it organizes the existing data in a very clear way.
This item is answered in the previous sentence.
The conclusions are scarce due to the low number of patients included in some of the studies. The authors point out that new and more complete studies on this subject are required.
The conclusions are in agreement with the findings.
The bibliographic citations are adequate.
The tables help to summarize and make the reading of the data clearer.
Depite this problem, I belive the study is of great interest and it can be published without changes
Author Response
Thank you very much for your comments.
We appreciate the time spent on the review. We have reviewed the English language and style.
Reviewer 2 Report
This is a well written paper in which recommendations were made for the diagnosis and treatment of influenza-associated invasive aspergillosis (IAPA) based on the available evidence and experience acquired in the management of patients with COVID-19-associated pulmonary aspergillosis (CAPA) by the CAPA/IAPA expert group.
Minor comments
Table 1. point 11: please replace Covid-19 with COVID-19
line 106: Please define DM
lines 106, 171, 209: please replace SARS-Cov2 -Cov with SARS- CoV-2
line 216: Microbiological methods/Recommendations (lines 133-134) are presented in section A. Presumamly, section B refers to radiological methods/recommendations. Please correct.
line 418: Please define PK/PD, DDI, TDM
line 490: please replace COVID with COVID-19
Author Response
Thank you very much for your comments.
We appreciate the time spent on the review. We have reviewed the English language and style.
In accordance with your recommendations we have modified the text of the manuscript according to your suggestions contemplated as minor comments. We are convinced that the manuscript will improve with your suggestions
Minor comments:
- We have replaced in Table 1. point 11 “Covid-19” with “COVID-19”
- In line 106 we have defined poorly controlled diabetes mellitus (DM)
- In revised versión we have replaced and corrected SARS- CoV-2 term.
- Thank you for the appreciation, we have corrected the error in the title of the section B refers to radiological methods/recommendations.
- We have corrected and defined PK/PD, DDI, TDM in teh text: “ Reconsider the antifungal therapy if pharmacological failure: pharmacokinetics (PK)/ pharmacodynamics (PD), drug-drug interaction (DDI) and therapeutic drug monitoring (TDM) is suspected.”
- COVID-19 has been replaced according with your suggestion.
Reviewer 3 Report
In the manuscript "Managing the Next Wave of Influenza and/or SARS-Cov-2 in 2 the ICU Practical Recommendations from an Expert Group", the authors establish practical recommendations for the diagnosis and treatment of influenza-associated invasive aspergillosis (IAPA), based on the available evidence and the experience gained in the management of patients with COVID-19-associated pulmonary aspergillosis (CAPA).
It is a well-written article and I think it will be helpful to clinicians in the management of patients with CAPA and IAPA. In addition, it is a guide that will optimize timely treatment for patients in order to resolve the disease. I only have one comment:
In question 2: The methods mentioned only identify the fungus at the genus level, however, the presence of "cryptic" species of Aspergillus, which are morphologically indistinguishable within the main sections of Aspergillus, has been evidenced in clinical samples, so it is important to identify these species in the clinical laboratory, due to the high frequency of isolates resistant to antifungals. It is known that species from sections Aspergillus, Circumdati, Clavati, Cremei, Nigri, Restricti, Usti, and Versicolores have been frequently isolated from the environment, while species from sections Fumigati, Terrei and Flavi have been isolated predominantly from clinical samples. Furthermore, several cryptic Aspergillus species have been reported as causative agents of invasive aspergillosis. In particular, A. quadrilineata, A. lentulus, A. alliaceus, A. tubingensis, A. calidoustus, A. viridinutans, A. udagawae, and A. felis have been reported to cause invasive aspergillosis, primarily in pulmonary infections. Therefore, I suggest that the authors mention the presence of these species, although, due to the apparent low frequency of cryptic Aspergillus species in clinical practice, so far, they are unlikely to influence the choice of empirical tests or the use of a primary antifungal.
References
I suggest reviewing the references carefully and following the style of the journal, since some references are not in a uniform format.
Line 34: Change “Aspergillus” to “Aspergillus”
Author Response
Thank you very much for your comments.
We appreciate the time spent on the review. We have reviewed the English language and style.
In accordance with your recommendations we have modified the text of the manuscript according to your suggestions. We are convinced that the manuscript will improve with your suggestions.
Thank you very much for your valuable appreciation of the impact that cryptic species of Aspergillus can have on the therapeutic management of aspergillosis. We added according to your recommendation a text mentioning it in the manuscript.
In question 2 we have added in the text: “ It is significan to contemplate in the etiology the possibility of isolation of cryptic species, efforts for their identification beyond the genus level are important since their profile of resistance to antifungal therapy.”
References has been reviewed according with the style of the journal.
We have changed according your suggestion “Aspergillus” to “Aspergillus”.